# Effects of Short-Term Lenvatinib Administration Prior to Transarterial Chemoembolization for Hepatocellular Carcinoma

**DOI:** 10.3390/cancers16091624

**Published:** 2024-04-23

**Authors:** Tetsuya Tachiiri, Kiyoyuki Minamiguchi, Ryosuke Taiji, Takeshi Sato, Shohei Toyoda, Takeshi Matsumoto, Yuto Chanoki, Hideki Kunichika, Satoshi Yamauchi, Sho Shimizu, Hideyuki Nishiofuku, Nagaaki Marugami, Yuki Tsuji, Tadashi Namisaki, Hitoshi Yoshiji, Toshihiro Tanaka

**Affiliations:** 1Department of Diagnostic and Interventional Radiology, Nara Medical University, Kashihara 634-8522, Japan; k135334@naramed-u.ac.jp (T.T.); kiyo829@naramed-u.ac.jp (K.M.); satotake@naramed-u.ac.jp (T.S.); s.toyoda@naramed-u.ac.jp (S.T.); t.matsumoto@naramed-u.ac.jp (T.M.); y.chanoki@naramed-u.ac.jp (Y.C.); k102972@naramed-u.ac.jp (H.K.); syamauchi@naramed-u.ac.jp (S.Y.); sho_shimizu.rad@naramed-u.ac.jp (S.S.); hmn@naramed-u.ac.jp (H.N.); marugami@naramed-u.ac.jp (N.M.); totanaka@naramed-u.ac.jp (T.T.); 2Department of Gastroenterology, Nara Medical University, Kashihara 634-8522, Japan; tsujih@naramed-u.ac.jp (Y.T.); tadashin@naramed-u.ac.jp (T.N.); yoshijih@naramed-u.ac.jp (H.Y.)

**Keywords:** hepatocellular carcinoma, lenvatinib, transarterial chemoembolization, combined therapy

## Abstract

**Simple Summary:**

Recently, favorable outcomes have been reported for hepatocellular carcinoma treated with transarterial chemoembolization (TACE) combined with lenvatinib (LEN-TACE). However, the optimal treatment protocol remains unclear. In this study, we performed a 4-day lenvatinib administration followed by TACE without an interval (short-term LEN-TACE). The objective was to assess changes in tumor hemodynamics following the 4-day lenvatinib administration and to evaluate the outcomes of this combined therapy. A significant decrease in intra-tumor flow after lenvatinib was observed, with a 100% technical success rate and no severe adverse events. Complete response rates (CR) at 1 month were 75%, and the 12-month progression-free survival (PFS) rate was 75.0%. The lipiodol-washout ratio between 1 week and 4 months after cTACE correlated with the arterial flow reduction radio by lenvatinib prior to TACE (*r* = −0.55). The short-term LEN-TACE is feasible and safe, demonstrating promising results, including a high CR rate and prolonged PFS.

**Abstract:**

Aim: Transarterial chemoembolization (TACE) combined with lenvatinib, employing a 4-day lenvatinib administration followed by TACE without an interval (short-term LEN-TACE), was performed for hepatocellular carcinoma (HCC). The aim was to assess tumor hemodynamics following the 4-day lenvatinib and to evaluate the treatment outcomes after the short-term LEN-TACE. Methods: 25 unresectable HCC patients received this combined therapy. Lenvatinib (4–12 mg) was administrated for 4 days prior to TACE. Perfusion CT scans were obtained before and after the lenvatinib administration. Either cTACE (76%) or DEB-TACE (24%) were performed. Results: intra-tumor blood flow significantly decreased after the 4-day lenvatinib (*p* < 0.05). The TACE procedure was successful with no severe adverse events in all patients. The overall complete response (CR) rate was 75% (cTACE 84%, DEB-TACE 40%). The lipiodol-washout ratio between 1 week and 4 months after cTACE correlated with the arterial flow reduction ratio by lenvatinib prior to TACE (*r* = −0.55). The 12-month progression-free survival (PFS) rate was 75.0%. Conclusions: The short-term LEN-TACE is feasible and safe, demonstrating promising outcomes with a high CR ratio, contributing to lipiodol retention in the tumor after cTACE, and extended PFS. To confirm the advantages of this treatment protocol, a prospective clinical trial is mandatory.

## 1. Introduction

Transarterial chemoembolization (TACE) has been widely used as the standard treatment for unresectable hepatocellular carcinoma (HCC) for 40 years [1]. Recent advancements in immunotherapies for HCC, such as atezolizumab/bevacizumab or durvalumab/tremelimumab, have prompted a shift in the role of TACE from a palliative to a curative treatment strategy [2,3]. The updated Barcelona Clinic Liver Cancer (BCLC) staging system highlights the indication of TACE for feasible selective access to feeding tumor arteries [4]. Within this context, TACE is designed to achieve a targeted survival duration of 2.5 years. Previously, there were reports indicating that achieving a complete response (CR) through TACE has been associated with prolonged overall survival [5]. The 2.5-year survival rate for patients achieving CR by TACE was approximately 80%, which significantly differed from that for non-CR patients who exhibited a survival rate of about 30%.

In recent years, combination therapy with molecular-targeted agents and TACE has garnered attention. The TACTICS trial demonstrated that the combination therapy of TACE and sorafenib significantly improved progression-free survival (PFS) compared to TACE monotherapy [6]. Subsequently, lenvatinib, a multi-kinase inhibitor targeting key receptors including vascular endothelial growth factor (VEGF) and fibroblast growth factor (FGF) receptors related to tumor angiogenesis, has changed the chemotherapy for HCC [7]. The REFLECT trial then illustrated an extension of PFS with lenvatinib compared to sorafenib. In a meta-analysis, lenvatinib demonstrates superior tumor responses and survival advantages compared to sorafenib when used as a first-line treatment for unresectable hepatocellular carcinoma, while maintaining a comparable incidence of adverse events [8]. Additionally, in the more recent TACTICS-L trial, the combination of lenvatinib and TACE demonstrated a high CR rate of 53.2% after TACE and a prolonged duration of response post-CR [9]. This rate appears remarkably higher when compared to the TACTICS trial, where the CR rate was 28% in the TACE plus sorafenib group. This achievement can be attributed to the synergistic effects of the combined approach involving TACE and lenvatinib. By inhibiting the function of VEGF and FGF receptors, lenvatinib effectively suppresses the angiogenesis of HCC [10,11]. Lenvatinib administration prior to TACE could normalize abnormal tumor vessels, thereby improving intraarterial drug distribution, and could also suppress the increase in VEGF induced by hypoxia from TACE [12,13]. In the TACTICS-L trial, lenvatinib was administered for 2–3 weeks, followed by a 2-day rest period before TACE. However, the optimal treatment protocol remains undetermined. Both the duration of lenvatinib administration and the rest period preceding TACE are anticipated to impact treatment outcomes. Prolonged lenvatinib administration may excessively eliminate tumor vasculature, leading to increased hypoxia and extensive tumor vessel pruning, thereby complicating drug delivery and resulting in TACE failure [14,15,16]. Additionally, long-term lenvatinib may prove intolerable for certain patients, hence a theoretically superior preference for short-term administration. A prior study revealed vascular normalization following a 4-day lenvatinib administration based on angiography [17]. Concerning the rest period before TACE, a few days’ pause in the administration of lenvatinib can decrease its anti-VEGF effects [18]. It is known that VEGF increases on the next day after TACE, which leads to poor prognosis [19].

Given these considerations, we attempted a 4-day lenvatinib administration followed by TACE without an interval (short-term LEN-TACE) for unresectable HCC, and retrospectively analyzed these results. The purpose of this study was to assess tumor hemodynamics following a 4-day lenvatinib administration and the outcomes associated with the short-term LEN-TACE.

## 2. Materials and Methods

### 2.1. Patient Data Collection and Eligibility Criteria

For the evaluation of the effects of the short-term lenvatinib, the databases at our institution were retrospectively reviewed. Our institutional review board approved this study, which was conducted ethically in accordance with the Declaration of Helsinki by the World Medical Association (IRB approval code No.3545). Informed consent was obtained from each patient before initiation of this combined therapy.

The study included cases of the short-term LEN-TACE performed at our institution from April 2022 to May 2023 for patients with unresectable HCC. Inclusion criteria were: (a) Diagnosis of HCC established based on findings from computed tomography (CT) and/or magnetic resonance imaging (MRI); (b) Unresectable HCC not suitable for surgical resection or percutaneous ablation therapy; (c) Child–Pugh A-B liver function; (d) Absence of significant arterioportal shunt formation through the tumor; (e) Absence of vascular invasion and extrahepatic spread.

### 2.2. LEN Administration before TACE

The administration duration of the lenvatinib was set at 4 days for all cases, and the dosage (ranging from 4 mg to 12 mg per day) was determined by the attending physician, considering factors such as the patient’s weight, age, and comorbidities. The administration was carried out once daily through oral intake.

### 2.3. Perfusion-CT

We performed CT with a contrast agent before and after lenvatinib administration prior to TACE. The CT scans were obtained within 4 weeks before lenvatinib administration and immediately before TACE by using a 320-slice area detector CT (Aquilion One, Canon Medical Systems Co., Ohtawara, Japan). The contrast injection protocol was as follows: 40 mL of non-ionic iodinated contrast agent (iopamidol 370 mg I/mL) was bolus-injected through the left antecubital vein at a rate of 5.0 mL/s. The CT scan, initiated simultaneously with injection, used the following acquisition parameters: tube voltage of 100 kV, tube current of 120 milliamperes, slice thickness of 3 mm, collimation of 320 × 0.5 mm, rotation time of 0.5 s, and cycle time of 2 to 4 s. Liver perfusion data were obtained using perfusion software available on postprocessing system (Body Perfusion, dual-input maximum slope analysis, Toshiba Medical Systems, Otawara, Japan), and calculations for arterial blood flow (ABF) of the tumors were performed. The tumors with a diameter < 15 mm, and/or with artifacts due to orthopedic metal, lipiodol nodules were excluded in this analysis.

### 2.4. TACE Procedure

The TACE procedures utilized a hybrid Angio-CT system (Angio-CT System, Infinix Activ; Canon Medical Systems Co., Ohtawara, Japan). After insertion of a 4 or 5 Fr catheter into the common hepatic artery origin, digital subtraction angiography (DSA) and CT during arterial injection of contrast material via the hepatic artery (CTHA) were obtained.

For the creation of a TACE navigation image, a 3D image of the hepatic artery was created using the arterial phase of CTHA, and tumors were extracted from the tumor phase of CTHA. Subsequently, a fusion image of these two phases was created using a 3D CT workstation (SYNAPSE VINCENT; Fujifilm, Tokyo, Japan). With the aid of the reconstructed navigation images, a 1.5 to 1.7 Fr tip microcatheter was selectively inserted into the tumor-feeding branches to minimize injection into non-targeted arteries.

The operator chose conventional TACE (cTACE) or drug-eluting beads TACE (DEB-TACE). In principle, cTACE was performed for small-diameter or few lesions, while DEB-TACE was performed for large tumors exceeding 6 cm in diameter or in cases where the patient was beyond up-to-11 criteria [20,21]. cTACE involved forming an emulsion of epirubicin solution and ethiodized oil (Lipiodol Ultra-Fluide; Guerbet, Villepinte, France) using a pumping emulsification technique with a glass membrane emulsification device (MicroMagic, Piolax Medical Device, Kawasaki, Japan). The emulsion was injected into the hepatic artery, followed by the injection of 1 mm gelatin sponge particles (Gelpart; Nippon Kayaku, Tokyo, Japan). DEB-TACE was performed by 100–300 μm DC Beads (Boston Scientific, Marlborough, MA, USA) loaded with epirubicin solution until the tumor staining disappeared.

### 2.5. Evaluation

#### 2.5.1. The Changes of Tumor Hemodynamics

We assessed changes in tumor hemodynamics following short-term lenvatinib administration. The ABF change ratio was calculated as follows;
$$ABF\;change\;ratio=\frac{ABFpost-ABFpre}{ABFpre}$$

Correlation between the ABF change ratio and the administered dosage of lenvatinib per body weight was evaluated.

In addition, the detectability of tumor stains on DSA after lenvatinib administration was evaluated. two radiologists (T.T. and K.M.) individually scored the degree of tumor staining on a 1–4 scale. Scores of 1–2 indicated poor depiction, while scores of 3–4 indicated good depiction. The tumor depiction degrees were assessed according to tumor diameters and the ABF change ratio.

#### 2.5.2. Technical Feasibility of short-Term LEN-TACE

The feasibility of the short-term LEN-TACE was evaluated. The success of superselective TACE was defined as follows: cTACE or DEB-TACE was performed via all tumor-feeding arteries, which were confirmed by selective CTHA via a microcatheter.

Adverse events (AEs) during the TACE procedures and the perioperative period within two weeks after TACE were evaluated. AEs were assessed using the National Cancer Institute’s Common Terminology Criteria for Adverse Events (CTCAE) version 5.0.

#### 2.5.3. Correlation between Lipiodol Washout in Tumors and Effects of Lenvatinib

The lipiodol-washout ratios at 1 week, 1 month and 4 months after cTACE were evaluated in patients who received cTACE without additional lenvatinib post-TACE, and also in those who received perfusion CT before and after TACE. The volume of lipiodol in a tumor was calculated by multiplying the tumor’s area (mm^2^) by its CT value (HU), following the methodology of previous studies [22,23]. The lipiodol volumes on CT at one month and four months after cTACE were compared with the lipiodol volume on CT at one week after cTACE. The Lipiodol washout ratio was defined using the formula below:$$Lipiodol-washout\;rate=\frac{\left(tumor^{\prime }s\;area\times CT\;value\;at\;1M\;or\;4M\right)-(tumor^{\prime }s\;area\times CT\;value\;at\;1W)}{\left(tumor^{\prime }s\;area\times CT\;value\;at\;1W\right)}$$

An investigation was carried out to explore the correlation between the lipiodol washout ratio and the ABF change ratio caused by lenvatinib prior to TACE.

#### 2.5.4. Tumor Response and Progression-Free Survival

Tumor response and PFS were evaluated according to RECICL [24]. The CR rate was assessed at 1 month after TACE. Some patients received lenvatinib administration at a dose of 4–8 mg after TACE, based on the physician’s consideration. PFS was defined as the time from the start of lenvatinib to the progression date. If a patient received alternative systemic chemotherapy or tracking was no longer possible before disease progression (PD), PFS was terminated at the last observation date. Median PFS, PFS rates at 6 months and 12 months were assessed.

### 2.6. Statistical Analysis

The statistical analysis was conducted using SPSS version 21.0 statistical software (SPSS Inc., Chicago, IL, USA). Clinical differences between cTACE and DEB-TACE were assessed using non-parametric Mann–Whitney U-Test and Pearson’s chi-square test, with significance set at *p* < 0.05. Differences in tumor blood flow before and after treatment were evaluated using a non-parametric Mann–Whitney U-Test. Pearson’s correlation test was employed to calculate correlation coefficients. All reported *p*-values are two-sided, with statistical significance defined as *p* < 0.05. The relationship between the ABF change ratio and the degree of tumor enhancement on DSA was examined using Pearson’s chi-square test, with significance set at *p* < 0.05. Median PFS was calculated using the Kaplan–Meier method and compared using the log-rank test. PFS rates at 6 and 12 months were similarly determined.

## 3. Results

### 3.1. Baseline Characteristics

There were 25 HCC patients who received short-term LEN-TACE therapy. Baseline patient characteristics are shown in Table 1. The median age was 80 years, and 72.0% of patients were male. The etiology of HCC was hepatitis C (28.0%) and hepatitis B (8.0%). Alcoholic and NASH were included in 16.0% and 20.0%, respectively. In total, 24.0% of the patients had no pre-existing liver disease. The Child–Pugh score was A in 88.0% of patients, modified ALBI (mALBI) score 1 or 2a in 64.0%, and alpha-fetoprotein < 200 ng/mL in 80.0% of patients [25]. In terms of liver cancer staging, 60% of cases were classified as early stage, and 40% as intermediate stage according to the APPLE consensus [26]. There were no patients in the advanced stage. The number of prior TACE treatments was 0 in 48.0% and 1–2 in 44.0%. Previous treatment other than TACE included 32.0% of patients with a history of surgical resection and 24.0% of patients with RFA. In the current LEN-TACE procedure, cTACE was chosen by the operator in 76% of cases, while DEB-TACE was selected in 24% of cases.

### 3.2. The Change of Tumor Hemodynamics

The changes in tumor hemodynamics were evaluated in 20 patients before and after lenvatinib administration prior to TACE, while the remaining 5 patients did not receive perfusion CT scans either before or after, or both assessments. A total of 20 out of 34 nodules in the 20 patients met the criteria for inclusion, 11 nodules were less than 15 mm in diameter, 2 nodules could not be assessed due to artifact caused by past treatment with lipiodol, and 1 case was excluded due to evaluation interference from artifact induced by orthopedic metallic implants. The mean arterial blood flow in the tumor prior to administration was 89.4 ± 36.1 mL/min/100 mL. The mean arterial blood flow after administration was 54.4 ± 22.3 mL/min/100 mL, indicating a consistent decrease in all cases (Figure 1). This difference was found to be statistically significant (*p* < 0.05 *). The correlation between the ABF change ratio and the dose of lenvatinib per body weight is shown in Figure 2. There was a significant correlation (*r* = −0.56, *p* < 0.05 *). Ten out of 20 nodules were judged as a poor depiction of DSA after the 4-day lenvatinib administration. Figure 3 demonstrates the detectability of each tumor based on tumor diameters and the ABF change ratio. The proportion of cases with poor depiction was significantly higher in the group with an ABF change ratio exceeding 40% (87.5%, 7/8) compared to those with an ABF change ratio of less than 40% (25.0%, 3/12) (*p* < 0.05 *).

### 3.3. Technical Feasibility and Safety of Short-Term LEN-TACE

Superselective TACE was successfully performed in all cases, achieving a 100% treatment success rate (Figure 4). There were no complications during the TACE procedures.

There were grade 1 hypertension, grade 1 hand-foot syndrome, and grade 1 anorexia were seen as AEs associated with 4-day lenvatinib administration prior to TACE. During the perioperative period within two weeks after TACE, 79% of patients had post-embolization syndrome, with grade 3+ elevations in aspartate aminotransferase (AST) and alanine aminotransferase (ALT), but they recovered, exhibiting no abnormalities within a week (Table 2).

#### 3.3.1. Correlation between Lipiodol Washout in Tumors and Effects of Lenvatinib

The lipiodol volume in tumors at 1 month and 1 week after TACE was compared in 11 nodules treated by cTACE without additional lenvatinib administration after TACE. A weak negative correlation (*r* = −0.30) between the lipiodol-washout ratio and ABF change ratio was observed. Furthermore, the lipiodol volume in tumors at 4 months and one week after TACE was compared in 9 nodules treated by cTACE without additional lenvatinib administration after TACE. In this case, a negative correlation (*r* = −0.55) was noted (see Figure 5 and Figure 6).

#### 3.3.2. Treatment Outcome of Short-Term LEN-TACE

One patient was lost to follow-up at 2 weeks after TACE due to the patient’s refusal of hospital admission. Among the remaining 24 patients, the median follow-up period was 10.6 months [range 1.9–18.0 months]. Eighteen patients achieved CR one month after TACE, resulting in a CR ratio of 75%. One patient died due to pneumonia at 3.7 months after TACE, while the remaining 23 patients are still surviving. Nine patients initiated additional lenvatinib treatment at 3.2 weeks (range: 1.6–7.2 weeks) after TACE. The swimmer plot of the 24 patients is shown in Figure 7. The overall 6-month PFS rate was 81.2% and the 12-month PFS rate was 75.0%, and at that point, the median PFS was not reached (Figure 8).

#### 3.3.3. Comparison of the Treatment Outcome between cTACE and DEB-TACE

Subgroup analyses were conducted for both cTACE and DEB-TACE. The background of patients who underwent cTACE and DEB-TACE is presented in Appendix A. There were no significant differences between the two groups in terms of age, weight, gender, etiology, Child–Pugh score, mALBI score, or AFP level. However, the tumor size was significantly larger in the DEB-TACE group, and the proportion of intermediate HCC cases in liver cancer staging was significantly higher in the DEB-TACE group compared to the cTACE group (*p* < 0.05 *).

In the cTACE group (*n* = 19), the CR rate at 1 month after TACE was 84%. The overall 6- and 12-month PFS rate was 88.2%. However, the DEB-TACE group (*n* = 5) had a 40% CR rate at 1 month after TACE, lower than the cTACE group (*p* < 0.05 *). The 6-month PFS was 40.0% and the median PFS was 5.9 months. There was no significant difference in PFS between the cTACE and DEB-TACE groups (log-rank test *p* = 0.079).

#### 3.3.4. Correlation between Clinical Characteristics and Treatment Outcome

Univariate analyses were conducted to evaluate the clinical characteristics (i.e., etiology) associated with the CR rate one month after TACE and PFS (Appendix A). However, no significant correlations were observed between these variables.

## 4. Discussion

In this study, it was observed that lenvatinib administration for 4 days prior to TACE resulted in an average 36% reduction in tumor intra-arterial blood flow. This effect consistently appeared in all cases. In our examination of clinical two cases previously reported, a comparison of angiography before and after lenvatinib administration for 4 days and 7 days showed decreased tumor arterial blood flows as well as the normalization of vascular structures, such as straightening of and less abnormal dilation of tumor vessel [17]. In addition, a mouse study conducted by Une et al. reported a reduction in tumor vascular density as well as an increase in blood vessels covered by pericytes and a reduction in intratumoral stromal pressure after 4 days of lenvatinib administration [27]. Therefore, the observed reduction in tumor intra-arterial blood flow obtained through perfusion CT in this study is consistent with previous research findings, indicating a reduction in arterial perfusion of the tumor due to normalization of tumor vasculature. Normalization of abnormal tumor vessels could enhance intra-arterial drug distribution, potentially resulting in a CR ratio [12].

The long-term administration of lenvatinib may induce tumor hypoxia and lead to the starvation of tumor vessels, potentially complicating TACE procedures [14,15,16]. Furthermore, prolonged use of lenvatinib may result in AEs such as fatigue and decreased appetite [28]. The TACTICS-L trial, in which lenvatinib was administered for 2–3 weeks prior to TACE, reported a lower complete response (CR) rate in patients with a relative dose intensity (RDI) when compared with those with 100% RDI [9]. This result could be interpreted as lenvatinib causing AEs during the 2–3 weeks, thereby interfering with the performance of an adequate TACE procedure. Therefore, short-term administration before TACE could be considered appropriate. In this study, lenvatinib administered prior to TACE resulted in only grade 1 AEs.

Short-term administration of lenvatinib allows patients to undergo TACE without interruption. It is well-known that an increase in serum VEGF levels 1–2 days after TACE is associated with shorter overall survival [18,19]. Lenvatinib prior to TACE could contribute to the suppression of the increase in VEGF. However, it is crucial to consider that lenvatinib has a half-life of 28 h, and its serum concentration becomes almost zero [29]. Therefore, uninterrupted TACE performed after lenvatinib administration is considered desirable.

TACE performed without an interval following lenvatinib is challenging due to poor visualization of tumor staining on DSA. In this study, 50% (10/20) of patients experienced inadequate tumor staining on DSA. Notably, patients with a decrease in the ABF change ratio of over 0.4 showed poor tumor staining on DSA in 87.5% (7/8) of cases. We utilized TACE navigation images, with the CTHA image reconstructed using a 3D-CT workstation, to identify tumor-feeding vessels during the TACE procedure. The application of this technique in our study facilitated the identification of tumor vessels in all cases, leading to superselective TACE procedures with a 100% technical success rate.

The therapeutic outcomes of short-term LEN-TACE in this study were favorable, demonstrating a CR rate of 75%. This rate is comparable, and even higher, than the reported CR rate of 53.2% at 4 weeks after TACE in TACTICS-L, despite the inclusion of many patients in the early stage. The PFS was also satisfactory. These results suggest that the 4-day lenvatinib administration followed by TACE without an interval may indeed yield sufficient synergistic effects. The combined therapy of TACE and systemic treatment is actively being pursued. Recently, the EMERALD-1 trial demonstrated prolonged PFS in TACE combined with durvalumab and bevacizumab compared to TACE alone [30]. The ongoing LEAP-012 trial, which combines lenvatinib with pembrolizumab and TACE, is underway [31]. Additionally, TALENTACE trials, which combine atezolizumab and bevacizumab with TACE, are currently in phase III [32]. The results of these trials are eagerly awaited. It will be necessary to compare the outcomes of these trials in the future.

Regarding the investigation of lipiodol retention after cTACE, the study observed the washout of lipiodol over time in many cases, and a decrease in washout corresponding to changes in intra-tumoral blood flow was also noted. Mechanisms for Lipiodol washout from the lesions have been suggested in previous reports [33,34], including insufficient embolization formation of arteries supplying the tumor, allowing residual blood flow to flush it out, and drainage through sinusoidal endothelial cells or lymphatic vessels such as Kupffer cells. In this study, the administration of lenvatinib resulted in the inhibition of abnormal vascular neoformation and normalization of tumor vasculature [10], suggesting a successful trans-arterial delivery of lipiodol. As a result, the arterial embolization supplying the tumor and the portal vein embolization through the tumor were effectively performed, and a washout block corresponding to the blood flow changes, believed to be influenced by the effects of lenvatinib, was observed in this study.

Regarding AEs, no serious complications were observed with lenvatinib prior to TACE, as mentioned above. AEs observed immediately after TACE treatment included liver function impairment, fever, abdominal pain, general malaise, etc., which were consistent with AEs reported in previous studies [35]. Although a temporary grade 3 or higher liver function impairment, considered as an embolism syndrome, occurred, prompt improvement was noted, and the safety profile of the treatment is considered to be well-tolerated.

The resumption of lenvatinib after TACE has not been definitively established at present. In this study, a substantial number of patients were in the early stage, and an assessment of the risk of recurrence based on the presence or absence of lenvatinib after TACE was not conducted. However, it is generally considered desirable to administer lenvatinib after TACE to prevent local, intrahepatic, and extrahepatic metastases in intermediate and advanced stage HCC. Previous reports demonstrated tumor numbers over 11 and beyond up-to-11 criteria were poor prognostic factors for TACE [36,37].

This study did not examine the drug metabolism of lenvatinib in HCC. However, we recognize the importance of future research efforts to describe several Quantitative Systems Pharmacology (QSP) models of HCC and computational models of liver regeneration [38]. In the future, we would like to integrate with these models to make predictions for LEN-TACE and similar treatment strategies.

This study has several significant limitations. Firstly, its retrospective design analyzes past data, which may contain biases and errors from the original data collection, potentially affecting the study’s results. Secondly, the small sample size poses limitations in generalizing the results to a broader population, as it reduces statistical power and increases the likelihood of false positive or negative findings. Additionally, data collection was confined to a single site, potentially leading to a patient population with distinct characteristics compared to other healthcare facilities. While all patients received lenvatinib before TACE, the use of lenvatinib after TACE varied among cases based on tumor status and comorbidities. This variability may impede the uniformity and comparability of results. Moreover, the short observation period limits a comprehensive assessment of long-term outcomes and side effects. This emphasizes the necessity for a more extended and extensive follow-up to gain a better understanding of the efficacy and safety of this treatment.

## 5. Conclusions

A 4-day administration of lenvatinib induces changes in tumor arterial blood flow, suggesting the potential for synergistic effects with vascular normalization. The 4-day lenvatinib administration followed by TACE without an interval is demonstrated as a feasible and safe treatment protocol. The outcomes of the short-term LEN-TACE are promising with a high CR ratio and long PFS. This study underscores the potential benefits of LEN-TACE and advocates for the necessity of prospective clinical trials.

## Figures and Tables

**Figure 1 cancers-16-01624-f001:**
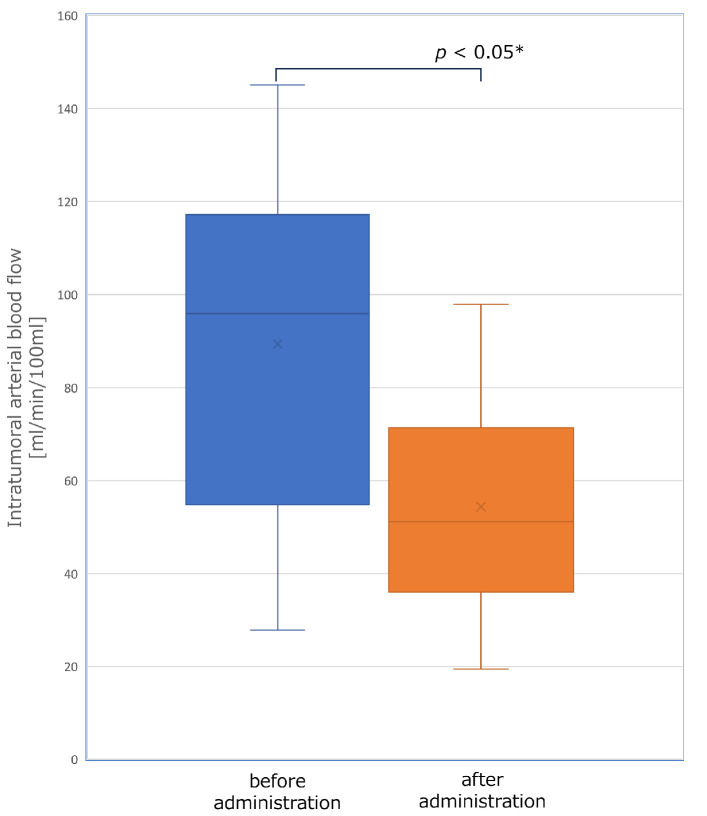
The change of intratumoral arterial blood flow following lenvatinib administration. There was a statistically significant reduction in the mean arterial blood flow in the tumor from 89.4 ± 36.1 mL/min/100 mL to 54.4 ± 22.3 mL/min/100 mL (*p* < 0.05 *).

**Figure 2 cancers-16-01624-f002:**
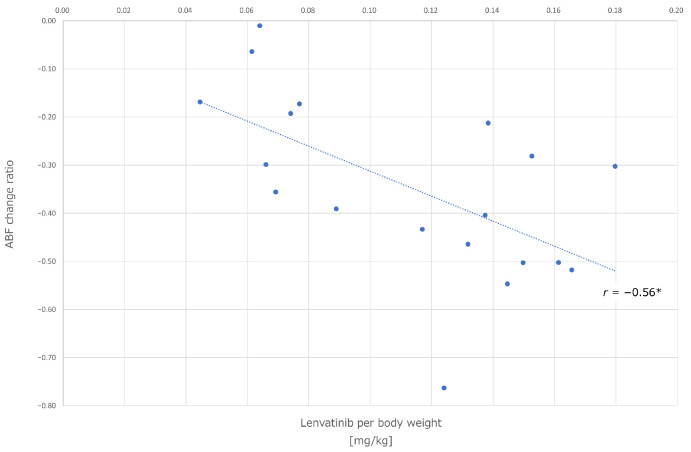
Correlation between the ABF change ratio and the dosage of lenvatinib per body. A negative correlation (*r* = −0.56 *) was observed between them. ABF, arterial blood flow.

**Figure 3 cancers-16-01624-f003:**
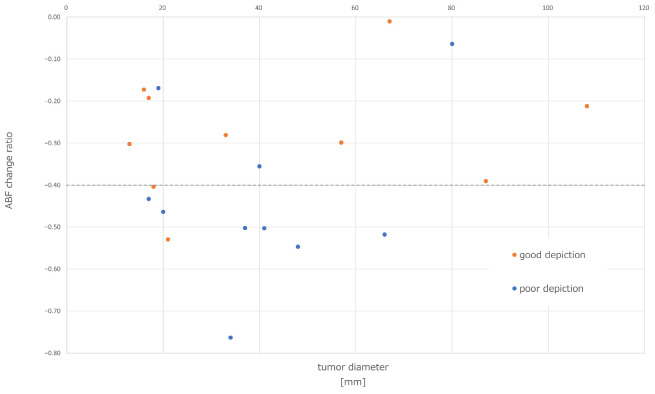
The detectability of tumor stains on DSA after lenvatinib administration. The detectability of tumor stains on DSA after lenvatinib administration was evaluated based on tumor diameters and the ABF change ratio. In total, 87.5% (7/8) of the cases with an ABF change ratio exceeding 40% were poor depiction, which was significantly higher than 25.0% (3/12) with an ABF change ratio of less than 40%. DSA, digital subtraction angiography; ABF, arterial blood flow.

**Figure 4 cancers-16-01624-f004:**
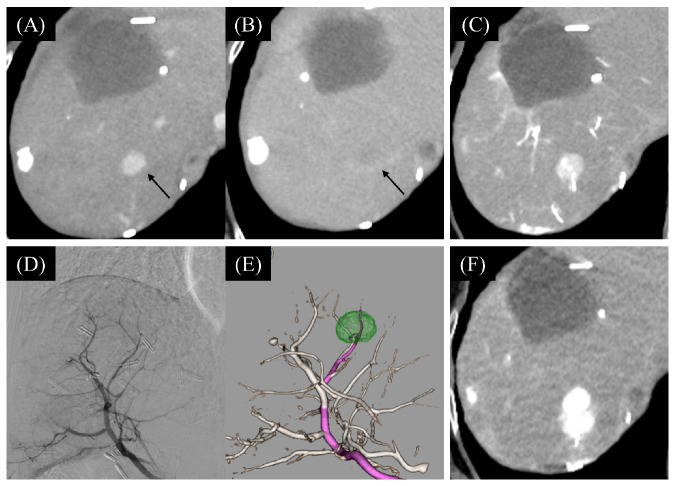
A case of the Sort-term LEN-TACE (superselective cTACE) performed by using a TACE navigation image. A 2.1 cm tumor (arrow) in segment 7 of the liver before lenvatinib administration showed early enhancement in the arterial phase (**A**) and washout in the delayed phase (**B**). During the TACE procedure after 4-day lenvatinib administration, the tumor was delineated on CTHA (**C**), but tumor staining was unclear on DSA (**D**). The TACE navigation images, with the tumor highlighted in green, reconstructed on a 3D-CT workstation (**E**), guided us in identifying and targeting the tumor-feeding arteries (pink) for a segmental TACE procedure. CT after the TACE (**F**) showed a good accumulation of lipiodol in the tumor. LEN, lenvatinib; TACE, transarterial chemoembolization; CTHA, computed tomography during hepatic arteriography; DSA, digital subtraction angiography.

**Figure 5 cancers-16-01624-f005:**
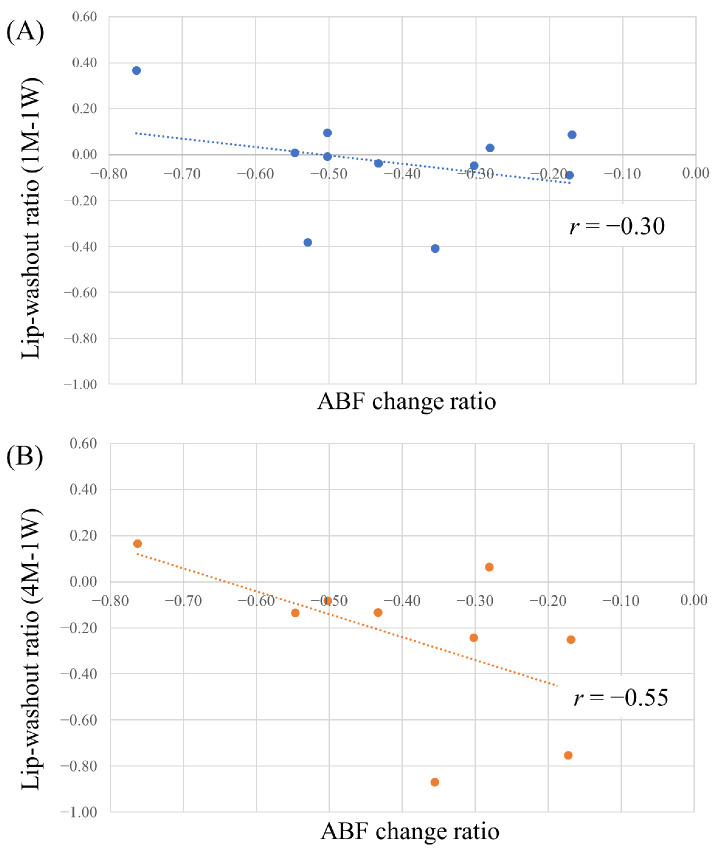
(**A**) Correlation between lipiodol-washout ratio (1M-1W) and the ABF change ratio; (**B**) between lipiodol-washout ratio (4M-1W) and the ABF change ratio. A higher reduction in tumor blood flow due to lenvatinib was observed to correlate with decreased lipiodol-washout, with correlation coefficients of *r* = −0.30 and *r* = −0.55. ABF, arterial blood flow.

**Figure 6 cancers-16-01624-f006:**
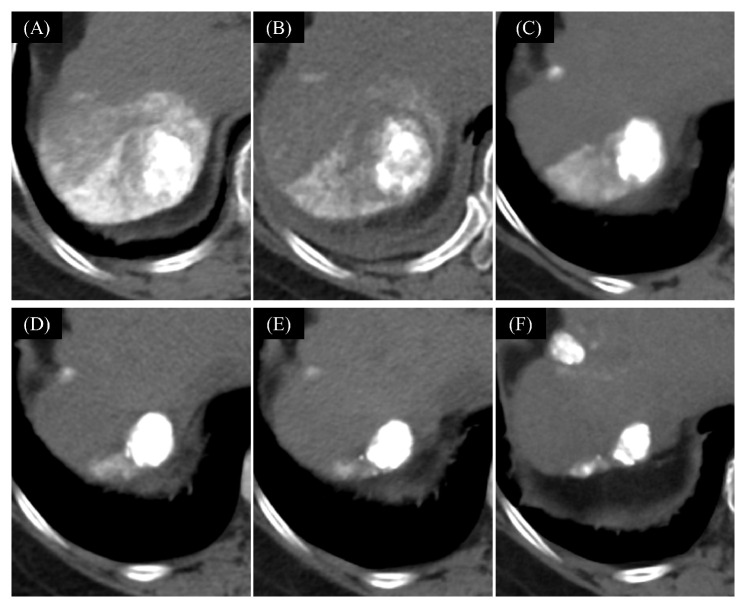
A case with good lipiodol retention after the short-term LEN-TACE (cTACE). The patient was a woman 70 s with a 3.4 cm HCC in the hepatic segment 7. A pre-TACE administration of 8 mg lenvatinib for 4 days was conducted. The intra-tumor blood flow changed from 140.4 mL/100 mg/min before to 33.3 mL/100 mg/min after, resultan ing in ABF change ratio of −0.76. CT immediately following cTACE (**A**) showed good lipiodol accumulation. One week after TACE (**B**), the tumor area was 565 mm^2^ with a mean CT value of 238 HU. After one month (**C**), the tumor area was 516.7 mm^2^ with a CT value of 356 HU, and the lipiodol-washout ratio (1M-1W) was +0.37. Four months post-TACE (**D**), the mean CT value was 358.4 mm^2^, the CT value was 438 HU, and the lipiodol-washout ratio (4M-1W) was +0.17. Observations at 6 months (**E**) and 9 months (**F**) after TACE also revealed continued good retention of lipiodol. LEN, lenvatinib; TACE, transarterial chemoembolization; HCC, hepatocellular carcinoma; ABF, arterial blood flow.

**Figure 7 cancers-16-01624-f007:**
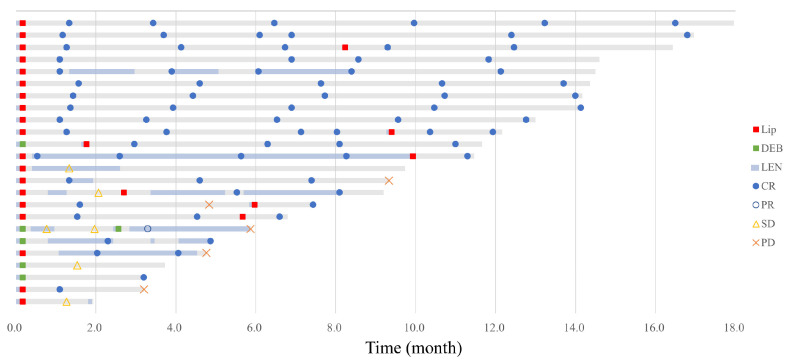
Swimmer plot of short-term LEN-TACE. TACE, transarterial chemoembolization; Lip, lipiodol-TACE; DEB, DEB-TACE; LEN, lenvatinib administration period; CR, complete response; PR, partial response; SD, stable disease; PD, progressive disease.

**Figure 8 cancers-16-01624-f008:**
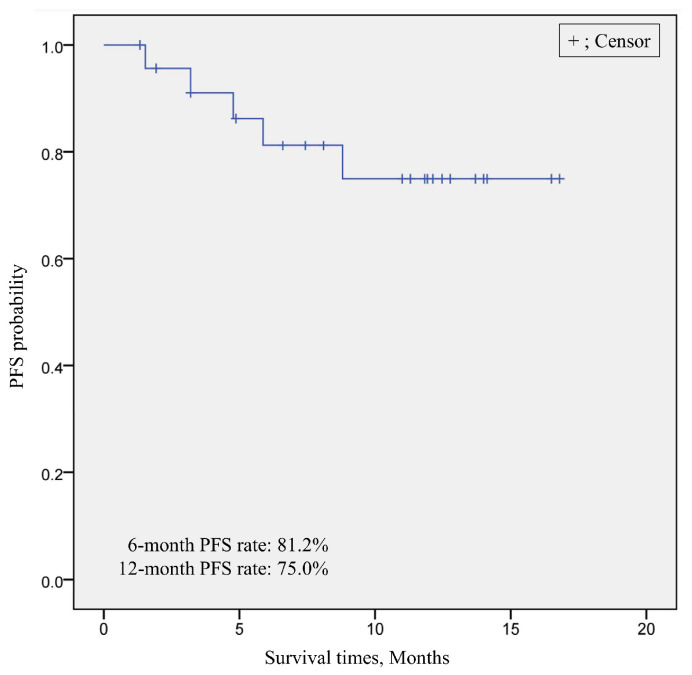
Kaplan–Meier survival curves of PFS according to RECICL. The overall 6-month PFS rate was at 81.2% and the 12-month PFS rate was 75.0%, and at that point, the median was not reached. RECICL, Response Evaluation Criteria in Cancer of the Liver; PFS, progression-free survival.

**Table 1 cancers-16-01624-t001:** Summary of clinical characteristics of 25 patients received the short-term LEN-TACE. mALBI, modified ALBI; AFP, α-fetoprotein; TACE, transarterial chemoembolization.

Characteristic	*n* = 25
Age, median (range), years	80 (63–90)
Weight, median (range), kg	58.2 (44.5–89.8)
Sex Male, *n* (%)	18 (72.0)
Etiology, %	
Hepatitis B	2 (8.0)
Hepatitis C	7 (28.0)
Non-B non-C	16 (64.0)
Child-Pugh stage, *n* (%)	
A	22 (88.0)
B	3 (12.0)
mALBI grade, *n* (%)	
1, 2a	16 (64.0)
2b	9 (36.0)
AFP, *n* (%)	
<200 ng/mL	20 (80.0)
≥200 ng/mL	5 (20.0)
Liver cancer staging, *n* (%)	
early	15 (60.0)
intermediate	10 (40.0)
TACE, *n* (%)	
cTACE	19 (76.0)
DEB-TACE	6 (24.0)

**Table 2 cancers-16-01624-t002:** Adverse events during the TACE procedures and the perioperative period. TACE, transarterial chemoembolization; Bil, bilirubin; AST, aspartate aminotransferase; ALT, alanine aminotransferase; ALP, alkaline phosphatase.

AEs, *n* (%)	Grade 1	Grade 2	Grade 3	Grade 4
pyrexia	6 (24)	0	0	0
malaise	8 (32)	1 (4)	0	0
appetite loss	8 (32)	0	0	0
nausea	4 (16)	0	0	0
abdominal pain	10 (40)	0	0	0
Dyspnea	2 (8)	0	0	0
Hypertension	0	1 (4)	0	0
Hypoalbuminemia	13 (52)	3 (12)	0	0
Bil increased	3 (12)	3 (12)	0	0
AST increased	3 (12)	3 (12)	12 (48)	7 (28)
ALT increased	6 (24)	6 (24)	11 (44)	0
ALP increased	1 (4)	0	0	0

## Data Availability

The datasets generated and/or analyzed during the current study are available from the corresponding author on reasonable request.

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
