# Peer review of "Effects of Short-Term Lenvatinib Administration Prior to Transarterial Chemoembolization for Hepatocellular Carcinoma"

_cancers, 2024, doi:10.3390/cancers16091624_

Round 1

Reviewer 1 Report (Previous Reviewer 2)

Comments and Suggestions for Authors

The authors consistently improved their manuscript based on my previous comments. Thank you!

Author Response

Thank you very much for your feedback. We greatly appreciate your acknowledgment of our efforts to improve the manuscript. Thank you again for your valuable input.

This manuscript is a resubmission of an earlier submission. The following is a list of the peer review reports and author responses from that submission.

Round 1

Reviewer 1 Report

Comments and Suggestions for Authors

This manuscript performs a retrospective study for a 4-day administration of lenvatinib followed by TACE without an interval for unresectable HCC. No severe adverse effect was observed and with significant decrease in intra-tumor flow. This is an interesting work with different dose strategy compared to ongoing clinical trial keeping in mind the vascular normalization window. I recommend accepting this manuscript with minor comments as follows:

(1)  Figure font is not legible please improve it

(2)  Why Lenvatinib and any other anti- angiogenic drugs such as bevacizumab, Ramucirumab etc?

(3)  Quantitative systems pharmacology (qsp) models can assist these retrospective study to understand the mechanism behind the treatments and synergy. QSP models are used extensively in Pharmaceutical field for dose strategy and clinical design. Authors should mention this as future scope, describing some qsp models for HCC or computational model for liver regeneration to be integrated with qsp model of hcc to make prediction for LEN-TACE and similar treatment strategy.

Reviewer 2 Report

Comments and Suggestions for Authors

Very interesting article. The limited sample size and the retrospective design represent major limitations and should be addressed as such in the Discussion.

Could the authors provide a subgroup comparative analysis between cTACE and DEB-TACE?

Do the authors have data also on other systemic agents in this setting?

THe authors should put their results more in the context of the current literature (for example mention and cite the recent SRMA: PMID: 34017396)

Did the etiology of the underlying cirrhosis influence the oncological outcomes?

Round 2

Reviewer 2 Report

Comments and Suggestions for Authors

The authors did not address any of my comments. The article was not improved so it cannot be accepted in the current form.